

# Age-related differences in flexibility in soccer players 8–19 years old

Antonio Cejudo[1], Francisco Javier Robles-Palazón[1], Francisco Ayala[2], Mark De Ste Croix[3], Enrique Ortega-Toro[1], Fernando Santonja-Medina[4] and Pilar Sainz de Baranda[1]

[1] Department Physical Activity and Sport/Faculty of Sport Sciences/Campus of Excellence Mare Nostrum, Universidad de Murcia, Murcia, Spain
[2] School of Physical Education, Faculty of Sport, Health and Social Care, University of Gloucestershire, Gloucester, United Kingdom
[3] School of Sport and Exercise, University of Gloucestershire, Gloucester, United Kingdom
[4] Virgen de la Arrixaca University Hospital/Faculty of Medicine/Campus of Excellence Mare Nostrum, Universidad de Murcia, Murcia, Spain

Corresponding author
Pilar Sainz de Baranda,
psainzdebaranda@um.es

## ABSTRACT

**Background**. Muscle flexibility is a main component of health-related fitness and one of the basic components of fitness for the performance in some sports. Sport and health professionals require the flexibility profile of soccer to define quantitative aims in the training of flexibility. The aim of this study was to identify age-related differences in lower extremity flexibility in youth soccer players.

**Methods**. Seventy-two young male soccer players (age: $13.0 \pm 3.1$ y; body mass: $50.5 \pm 15.3$ kg; stature $158.2 \pm 16.8$ cm; BMI: $19.6 \pm 2.6$ kg/m$^2$) completed this study. Measures of eleven passive hip (hip extension (HE), hip adduction with hip flexed 90° (HAD-HF90°), hip flexion with knee flexed (HF-KF) and extended (HF-KE), hip abduction with hip neutral (HAB) and hip flexed 90° (HAB-HF90°), hip external (HER) and internal (HIR) rotation), knee (knee flexion (KF)) and ankle dorsiflexion (ankle dorsiflexion with knee flexed (ADF-KF) and extended (ADF-KE)) ranges of motion (ROM) were taken. Descriptive statistics were calculated for hip, knee and ankle ROM measured separately by leg (dominant and non-dominant) and age-group (U10, U12, U14, U16 and U19). The data was analysed using a one-way analysis of variance (ANOVA) to examine the interaction of 11 ROM in the different players' age-group.

**Results**. Generally, U10 and/or U12 soccer players obtain the highest mean value in almost all ROM evaluated (U10: HAD-HF [$39.6° \pm 4.3°$], ADF-KE [$32.3° \pm 4.1°$], HER [$63.5° \pm 5.6°$] and HAB-HF90° [$64.1° \pm 7.5°$]; U12: HE [$17.7° \pm 6.2°$], HAB [$35.6° \pm 3.0$], HIR [$60.8° \pm 4.7°$] and KF [$133.8° \pm 7.1°$]). Nonetheless, significant differences between the players' age-groups are just found in HAD-HF90° ($p = .042$; ES = .136), HAB ($p = .001$; ES = .252), HIR ($p = .001$; ES = .251), HER ($p < .001$; ES = .321) and HAB-HF90° ($p < .001$; ES = .376) ROM, showing a progressive and irregular decrease in these ROM until the U19 team.

**Conclusion**. The findings of this study reinforce the necessity of prescribing exercises aimed at improving HAD-HF90° ROM in U16, HAB ROM in U14, HIR ROM in U16 and U19, HER ROM in U12 and U19, and HAB-HF90° ROM in U16 and U19 players within everyday soccer training routines.

# INTRODUCTION

Soccer is by far the world's most popular sport (*Dvorak, Junge & Graf-Baumann, 2004*). According to the survey conducted by the International Federation of Association Football (FIFA) in 2006, more than 270 million participants played soccer in the world and most of them were male players (90% of all registered players), with younger soccer players comprising the greatest proportion (54.7%) (*FIFA, 2006*).

Soccer requires players to perform many repeated high intensity movements such as sudden acceleration and deceleration, lots of changes of direction, jumping and landing tasks, as well as many situations in which players are involved in tackling to keep or obtain possession of the ball (*Krustrup et al., 2010*). Optimal performance in these actions depends upon a variety of anthropometrical and physiological properties (*Arnason et al., 2004*; *Pion et al., 2015*; *Stolen et al., 2005*). For example, body composition has been identified as an important factor which could adjust the performance in soccer players across the season due to the contribution that body fat and lean muscle mass have on some physical abilities (*Carling & Orhant, 2010*; *Milanese et al., 2015*). Likewise, some authors identified endurance, repeated-sprint ability, velocity, agility and strength as the main properties and optimal performance factors in this sport (*Haugen, Tonnessen & Seiler, 2013*; *Rebelo et al., 2014*; *Stolen et al., 2005*; *Wong et al., 2015*). Nevertheless, it seems anthropometrical and physiological demands are specific for each soccer player and depend on the participants' sex, position and age (*Di Salvo et al., 2007*; *Le Gall et al., 2010*; *Oyón et al., 2016*; *Wong et al., 2009*).

Muscle flexibility is a main component of health-related fitness, and one of the basic components of the performance in some sports (*Kraemer & Gómez, 2001*); in soccer, deficits in some ranges of motion might restrict specific technical skills and reduce players' performance (*García-Pinillos et al., 2015*; *Mills et al., 2015*; *Nunome et al., 2006*). Although there is no consistent scientific evidence about the relationship between flexibility and injury risk, it seems lower range of motion values in soccer players could also increase the risk of some muscle injuries (*Bradley & Portas, 2007*; *Henderson, Barnes & Portas, 2010*; *Witvrouw et al., 2003*). In sport, it has been observed that flexibility is subject to sex (*Gómez-Landero, Vernetta & López-Bedoya, 2013*; *Kibler & Chandler, 2003*), tactical position (*Oberg et al., 1984*; *Sporis et al., 2011*), dominant laterality (*Bittencourt et al., 2014*; *Rahnama, Lees & Bambaecichi, 2005*) and competitive level (*Rubini, Costa & Gomes, 2007*). However, there are few articles where muscle flexibility was directly related to the age of soccer players (*Malina et al., 2007*; *Manning & Hudson, 2009*; *Nikolaïdis, 2012*; *Vaeyens et al., 2006*). A general trend towards the flexibility reduction over ages has been reported by previous studies conducted in non-athlete population (*McKay et al., 2017*). The knowledge of flexibility changes in relation to the age of soccer players could show the variation of this capability throughout the different phases of sport specialization, providing useful information about the critical flexibility stages and the primary affected muscles to physical trainers. Therefore, the aim of this study is to identify age-related differences in lower extremity flexibility in young soccer players.

**Table 1  Demographic variables of the players of a soccer club (mean ± standard deviation).**

|  | U10 (n = 16) | U12 (n = 15) | U14 (n = 13) | U16 (n = 15) | U19 (n = 13) | Total (n = 72) |
|---|---|---|---|---|---|---|
| Age (years) | 8.9 ± 0.9 | 11.6 ± 0.5 | 12.7 ± .7 | 14.9 ± 0.7 | 17.7 ± .8 | 13.0 ± 3.1 |
| Body mass (kg) | 33.5 ± 6.3 | 39.7 ± 6.4 | 52.1 ± 8.8 | 62.0 ± 9.1 | 70.2 ± 5.2 | 50.5 ± 15.3 |
| Height (cm) | 136.4 ± 6.7 | 149.3 ± 8.3 | 161.0 ± 8.7 | 172.2 ± 7.7 | 176.9 ± 5.5 | 158.2 ± 16.8 |
| BMI (kg/m$^2$) | 17.9 ± 2.21 | 17.6 ± 1.1 | 19.7 ± 2.3 | 20.5 ± 1.8 | 22.6 ± 1.8 | 19.6 ± 2.6 |

**Notes.**

BMI, body mass index.

# MATERIALS & METHODS

## Participants

Seventy-two young soccer players completed this study. The participants were recruited from five different teams (U10 = 16 players; U12 = 15 players; U14 = 13 players; U16 = 15 players; U19 = 13 players) of a youth soccer academy (Table 1). All the participants were outfield players and participated regularly in sport (3–4 training sessions and 1 match per week). Also, none of the participants were involved in systematic and specific stretching regimes in the last 6 months, apart from the 1–2 sets of 8–10 s of static stretches designated for the major muscles of the lower extremities (e.g., hamstrings, quadriceps, adductors and triceps surae) that were performed daily during their pre-exercise warm-up and post-exercise cool down phases.

The exclusion criterion was history of orthopaedic problems to the knee, thigh, hip, or lower back in the last 3 months due to the fact that residual symptoms could have an impact in the habitual players' movement competency and/or lower extremity ROM profile. The study was conducted at the end of the preseason phase of the year 2016. The time frame of the study was selected to make sure that the players recruited to each team were definitive and stable within the testing period.

Before any participation, experimental procedures and potential risks were fully explained to both parents and children in verbal and written form, and written informed consent was obtained. The experimental procedures used in this study were in accordance with the Declaration of Helsinki and were approved by the Ethics and Scientific Committee of the University of Murcia (Spain) (ID: 1551/2017).

## Testing procedure

The passive hip extension (HE), hip adduction with hip flexed 90° (HAD-HF90°), hip flexion with knee flexed (HF-KF) and extended (HF-KE), hip abduction with hip neutral (HAB) and hip flexed 90° (HAB-HF90°), hip external (HER) and internal (HIR) rotation, knee flexion (KF), ankle dorsiflexion with knee flexed (ADF-KF) and extended (ADF-KE) ROM of the dominant and non-dominant leg were assessed following the methodology previously described (*Cejudo et al., 2014a*) (Fig. 1).

These tests were selected because they have been considered appropriate by American Medical Organizations, *American Academy of Orthopedic Association (1965)* and American Medical Association (*Gerhardt, Cocchiarella & Lea, 2002*) and included in manuals of sports medicine and science (*Magee, 2002*; *Palmer & Epler, 2002*) based on reliability and

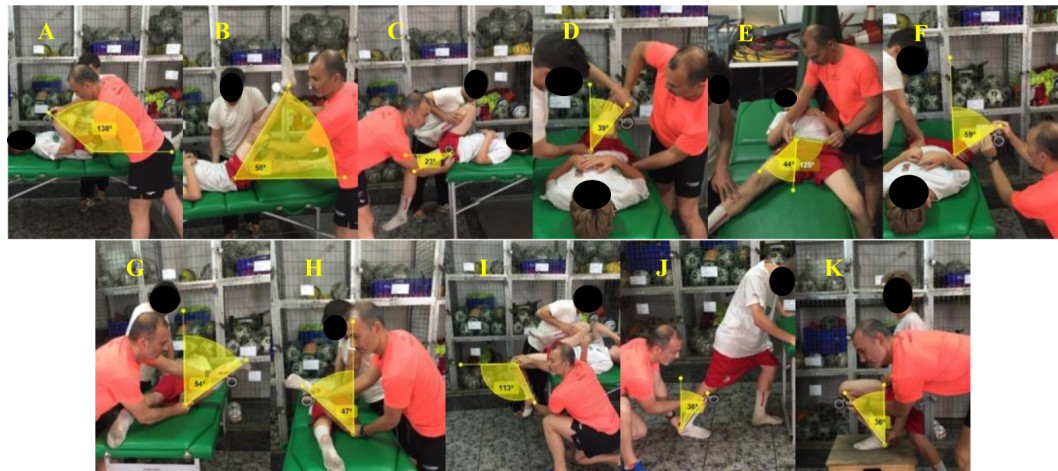

**Figure 1 Lower extremity ranges of motion of the "ROM-SPORT" protocol.** (A) Hip flexion with knee flexed test (HF-KF); (B) Hip flexion with knee extended test (HF-KE); (C) Hip extension test (HE); (D) Hip adduction with hip flexed 90° test (HAD-HF90°); (E) hip abduction with hip neutral test (HAB); (F) hip abduction with hip flexed 90° test (HAB-HF90°); (G) Hip internal rotation test (HIR); (H) Hip external rotation test (HER); (I) Knee flexion test (KF); (J) Ankle dorsiflexion with knee extended test (ADF-KE); (K) Ankle dorsiflexion with knee flexed test (ADF-KF).

validity studies, anatomical knowledge, and extensive clinical and sport experience (*Cejudo et al., 2015a*; *Cejudo et al., 2015b*; *Cejudo et al., 2014b*). In addition, the intra-operator variability was analysed for each muscle flexibility measure using a test-retest design. Before data collection, the reliability coefficient was evaluated on 20 healthy athletes. The range of motion was measured twice with a 2-week interval. An interclass correlation coefficient (ICC) and the minimal detectable change at 95% confidence interval (MDC$_{95}$) were calculated from the results of subsequent measurements. Results of pre-and-post-measurements showed a high reliability coefficient in all the tests (HF-KF [0.94], HF-KE [0.97], HE [0.97], HAD-HF [0.97], HAB [0.95], HAB-HF90° [0.96], HER [0.96], HIR [0.96], KF [0.95], ADF-KE [0.95], and ADF-KF [0.95]). The MDC$_{95}$ for each flexibility measure ranged from 3.7° to 6.9° (HF-KF [6.2°], HF-KE [6.1°], HE [3.7°], HAD-HF [5°], HAB [5.5°], HAB-HF90° [4.7°], HER [4.7°], HIR [4.1°], KF [6.9°], ADF-KE [4.7°], and ADF-KF [5°]).

One week before the start of the study, all the soccer players completed a familiarization session with the purpose of getting to know the correct technical execution of the exploratory tests by means of the practical realization of each one of them. The dominant leg was defined as the participant's preferred kicking leg. All tests were carried out by the same two sport scientists (one conducted the tests and the other ensured proper testing position of the participants throughout the assessment manoeuvre) under stable environmental conditions. The sport scientists were blinded to the purpose of the study and to the test results from previous testing sessions.

Prior to the testing session, all participants performed the dynamic warm-up designed by *Taylor et al. (2009)*. The overall duration of the entire warm-up was approximately 20 min.

A 3–5 min rest interval between the end of the warm-up and beginning of the ROM assessment was given to the soccer players because in a pilot study with 10 participants of similar age and training status, sometime was required for practical reasons, like rehydrating and drying their sweat prior to the ROM assessment. More importantly, it has been shown that the effects elicited by the dynamic warm-up on muscle properties might last more than 5 min (*Ayala et al., 2016*) and hence, decreases in ROM values within the 3–5 min rest interval were not expected.

After the warm-up, soccer players were instructed to perform, in a randomised order, two maximal trials of each ROM test for each leg, and the mean score for each test was used in the analysis. Soccer players were examined wearing sports clothes and without shoes. A 30s rest was given between trials, legs and tests.

For the measurement, an ISOMED Unilevel inclinometer (Portland, Oregon) was used with an extendable telescopic rod (*Gerhardt, Cocchiarella & Lea, 2002*), a metal goniometer with long arm (Baseline® Stainless) and "lumbosant" -lumbar support- to standardize the lumbar curvature (*Santonja, Ferrer & Martínez, 1995*; *Sainz de Baranda et al., 2014*).

Before each assessment session, the inclinometer was calibrated to 0° with either the vertical or horizontal axis. The angle between the longitudinal axis of the mobilized segment was recorded (following its bisector) with the vertical or the horizontal (*Cejudo et al., 2014a*; *Gerhardt, Cocchiarella & Lea, 2002*). Regarding the assessment of hip abduction movement, a metal goniometer of long arm (Baseline® Stainless) was used.

One or both of the following criteria determined the endpoint for each test: (a) an examiner palpable or appreciated some compensation movement that increased the ROM onset of pelvic rotation, and/or (b) the soccer player feeling a strong but tolerable stretch, slightly before the occurrence of pain (*Cejudo et al., 2014b*).

### Statistical analysis

Prior to the statistical analysis, the distribution of raw data sets was checked using the Kolmogorov–Smirnov test and demonstrated that all data had a normal distribution ($p > .05$). Descriptive statistics including means and standard deviations were calculated for hip, knee and ankle ROM measurements, separately by leg (dominant and non-dominant). Dependent sample t-tests were carried out to assess differences between the values of the dominant and non-dominant sides. Also, data was analysed using a one-way analysis of variance (ANOVA) to examine the interaction of 11 ROM at different youth teams. Post hoc comparisons were made using the Bonferroni test for pair wise comparisons. Analysis was completed using SPSS version 20 (SPSS Inc, Chicago, IL, USA). The effect sizes (ES) of each variable were tested using eta squared ($\eta2$) between groups (.01 = small effect, .06 = medium effect, and .14 = large effect (*Cohen, 1988*). Statistical significance was set at $p < .05$.

## RESULTS

Statistical analysis reported no differences between dominant and non-dominant sides for each ROM value, so the mean scores were used for comparing age-groups. Table 2 shows the results of the 11 ROM variables of the ROM-SPORT protocol in soccer players,

**Table 2  Outfield based players' descriptive values (mean ± SD) for 11 passive ranges of motion in the five categories analyzed ($n = 72$).**

| Range of motion (grades) | U10 ($n = 16$) | U12 ($n = 15$) | U14 ($n = 13$) | U16 ($n = 15$) | U19 ($n = 13$) | Total ($n = 72$) |
|---|---|---|---|---|---|---|
| HE | 16.8° ± 8.1° | 17.7° ± 6.2° | 12.4° ± 4.9° | 12.3° ± 8.1° | 11.4° ± 6.0° | 14.3° ± 7.2° |
| HAD-HF90° | 39.6° ± 4.3°[d] | 38.1° ± 4.1° | 38.4° ± 3.2° | 34.9° ± 5.4°[a] | 36.2° ± 4.8° | 37.5° ± 4.6° |
| ADF-KE | 32.3° ± 4.1° | 29.7° ± 3.8° | 31.3° ± 3.3° | 31.6° ± 5.1° | 30.6° ± 3.9° | 31.1° ± 4.1° |
| ADF-KF | 36.4° ± 4.0° | 34.8° ± 4.1 | 36.5° ± 4.6° | 35.8° ± 4.8° | 34.6° ± 4.2° | 35.6° ± 4.3° |
| HAB | 34.5° ± 3.7°[c] | 35.6° ± 3.0°[c, d] | 29.6° ± 4.9°[a, b, e] | 31.5° ± 4.5°[b] | 34.4° ± 2.9°[c] | 33.2° ± 4.4° |
| HIR | 55.7° ± 8.5° | 60.8° ± 4.7°[d, e] | 55.4° ± 7.7° | 49.5° ± 8.1°[b] | 49.6° ± 8.4°[b] | 54.3° ± 8.5° |
| HER | 63.5° ± 5.6°[b, e] | 50.1° ± 4.8°[a, d] | 55.7° ± 12.7° | 61.2° ± 5.6°[b, e] | 50.4° ± 11.1°[a, d] | 56.4° ± 9.8° |
| HAB-HF90° | 64.1° ± 7.5°[c, d, e] | 62.8° ± 4.8°[c, d, e] | 55.3° ± 5.4°[a, b] | 53.9° ± 6.1°[a, b] | 53.3° ± 6.8°[a, b] | 58.1° ± 7.6° |
| HF-KE | 70.0° ± 9.8° | 69.7° ± 7.8° | 70.3° ± 8.5° | 73.4° ± 9.9° | 74.4° ± 8.2° | 71.5° ± 8.9° |
| KF | 130.8° ± 15.1° | 133.8° ± 7.1° | 127.0° ± 9.1° | 124.8° ± 10.6° | 120.4° ± 16.4° | 127.6° ± 12.7° |
| HF-KF | 136.8° ± 9.1° | 135.2° ± 5.6° | 136.3° ± 3.7° | 131.4° ± 6.8° | 136.9° ± 7.3° | 135.3° ± 7.0° |

Notes.
[a] Significantly different from U10 ($p < .05$).
[b] Significantly different from U12 ($p < .05$).
[c] Significantly different from U14 ($p < .05$).
[d] Significantly different from U16 ($p < .05$).
[e] Significantly different from U19 ($p < .05$).

HE, hip extension test; HAD-HF90°, hip adduction with hip flexed 90° extended test; ADF-KE, ankle dorsiflexion with knee extended test; ADF-KF, ankle dorsiflexion with knee flexed test; HAB, hip abduction test; HIR, hip internal rotation test; HER, hip external rotation test; HAB-HF90°, hip abduction with hip flexed 90° test; HF-KE, hip flexion with knee extended test; KF, knee flexion test; HF-KF, hip flexion with knee flexed test.

differentiating the data between the 5 youth teams. Generally, U10 and/or U12 soccer players obtained the highest mean value in almost all ROM evaluated (HE, HAD-HF, ADF-KE, HAB, HIR, HER, HAB-HF90° and KF). Nonetheless, significant differences between the players' age-group (large effect) are only found in HAD-HF90° ($F_{4,67} = 2.629$; $p = .042$; ES $= .136$), HAB ($F_{4,67} = 5.642$; $p = .001$; ES $= .252$), HIR ($F_{4,67} = 5.624$; $p = .001$; ES $= 0.251$), HER ($F_{4,67} = 7.930$; $p < .001$; ES $= .321$) and HAB-HF90° ($F_{4,67} = 10.074$; $p < .001$; ES $= .376$) ROM, showing a progressive and irregular decrease in these ROM until the U19 team. On the contrary, the greatest mean value in the HF-KE ($F_{4,67} = .847$; $p = .501$; ES $= .048$) and HF-KF ($F_{4,67} = 1.674$; $p = .166$; ES $= .091$) ROM is observed in U19 players, but these differences are not statistically significant.

U10 players report the highest values in HAD-HF90°, HER and HAB-HF90° ROM. The Bonferroni post hoc comparisons reflect statistical differences with U16 players ($p = .035$) in HAD-HF90°, U12 ($p < .001$) and U19 ($p = .001$) players in HER, and U14 ($p = .003$), U16 ($p < .001$) and U19 ($p < .001$) players in HAB-HF90°. U12 players obtained the highest ROM in HAB and HIR; the Bonferroni post hoc comparisons showed statistical differences with U14 ($p = .001$) and U16 ($p = .04$) in HAB, and U16 ($p = .001$) and U19 ($p = .003$) in HIR. The Bonferroni post hoc test also indicated significant differences between U12 players and U14 ($p = .019$), U16 ($p = .002$) and U19 ($p = .001$) soccer players in HAB-HF90° ROM. Finally, the Bonferroni post hoc test showed significant differences in U14 vs. U10 ($p = .011$) and U14 vs. U19 ($p = .021$) in HAB, and U16 vs. U12 ($p = .005$) and U16 vs. U19 ($p = 0.01$) soccer players in HER (Fig. 2).

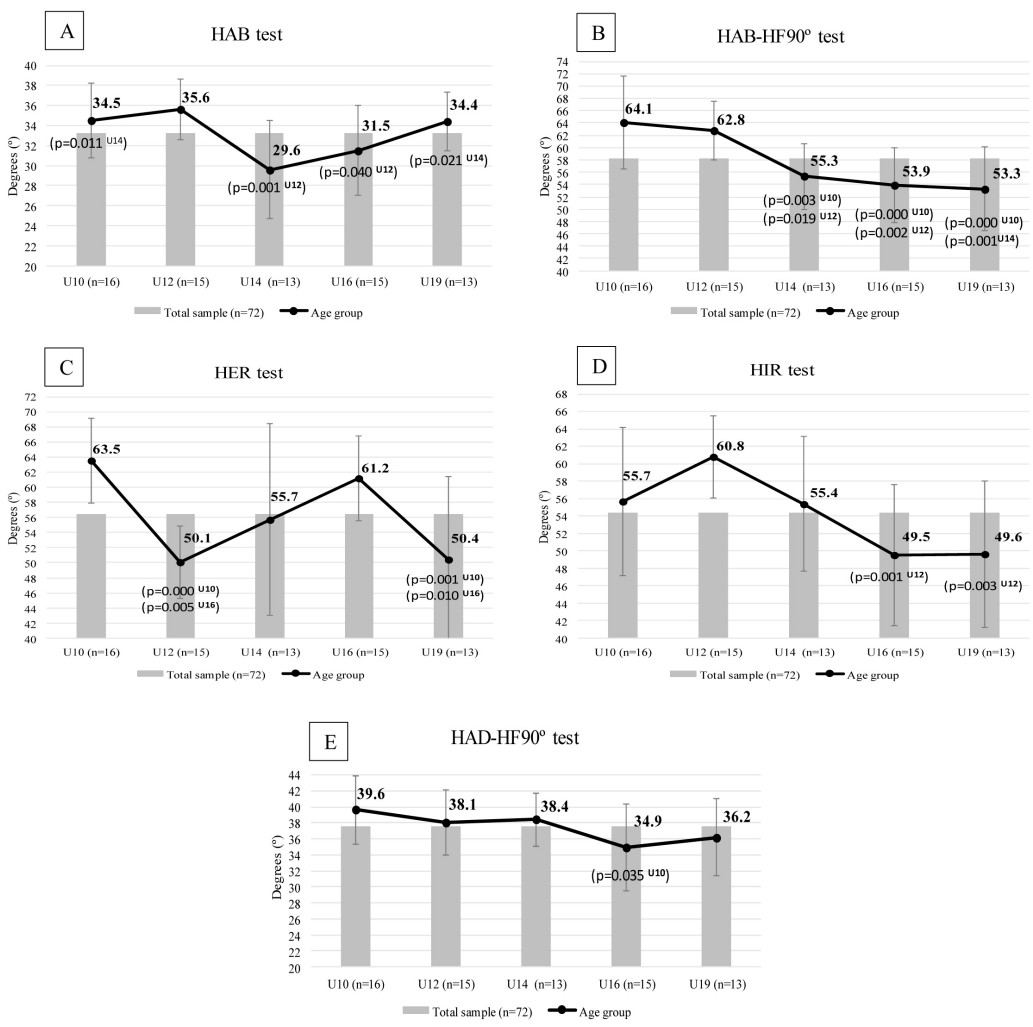

**Figure 2  Age-related demands in flexibility in soccer players 8–19 years old.** Statistically significant differences between age group ($p \leq .040$) with U10, U12, U14, U16, U19. (A) HAB: hip abduction with hip neutral test; (B) HAB-HF90°: hip abduction with hip flexed 90° test; (C) HER: hip external rotation test; (D) HIR: hip internal rotation test; (E) HAD-HF90°: hip adduction with hip flexed 90° test.

## DISCUSSION

The purpose of the current study was to identify age-related differences in lower extremity flexibility in young soccer players. The principal results of this research indicated that the U10 and U12 soccer players had the highest values in several (HE, HAD-HF, ADF-KE, HAB, HIR, HER, HAB-HF90° and KF), but not all (HF-KE and HF-KF) of the ROM assessed in comparison with their counterpart older players.

Flexibility seems to decline with age (*Medeiros, Araújo & Araújo, 2013*). *Johns & Wright (1962)* determined that the relative contributions of soft tissue to total resistance encountered at a joint are as follows: joint capsules, 47%; muscle and its fascia, 41%; tendons and ligaments, 10%; and skin, 2%. Evidence suggests that biological changes such

as tendon stiffening, joint capsule changes, or muscle changes could be responsible factors for the age-related decrease in flexibility (*Adams, O'shea & O'shea, 1999*). In this sense, our data supports this statement showing a slight tendency to reduce the ROM during different periods of soccer practice (specially in HE, HIR, HAB-HF90° and HAD-HF90° ROM (Fig. 2)). A similar pattern in some of the hip ROM over the development phases was previously reported by *Manning & Hudson (2009)* who also found significant lower ROM values in HE, HIR, HAB-HF90° in senior soccer players compared to their equals' young players. The poor flexibility showed by older players in these ROM may also reflect an adaptive response to playing soccer of soft tissue around the joints that improves stability at the specific joint, or lack of attention to flexibility practices in training (*Ostojić & Stojanović, 2007*). Thus, training and repetition of specific soccer skills like kicking the ball might explain the reduction of scores in HIR, HE and HAB-HF90o tests throughout the different age-groups as a result of adaptation to this continuous contraction of external hip rotator, hip flexor and hip adductor muscles. Indeed, these kicking adaptations would be in agreement with the lower values found by *López-Valenciano et al. (2019)* in his recent study in professional soccer players in HE (8.9°) and HIR (47.1°) ROM. Taking into account that these reductions could predispose players to abnormal movement patterns and increase the degeneration or capsular tightening in the hip joint (*Manning & Hudson, 2009*), it is essential to perform stretching exercises aiming to improve the hip ROM.

On the contrary, older players have shown better values of ROM in HF-KF and, mainly, in HF-KE. The HF-KE test assesses the flexibility of hamstring muscles. Some previous studies have also shown similar increases in older soccer players' hamstring flexibility in comparison with their younger counterparts using the sit-and-reach test (*Nikolaïdis, 2012*; *Vaeyens et al., 2006*). During the early childhood and adolescence ages the growth of bone and muscle play important roles in force development, musculoskeletal loading and motor control during childhood. Differential bone growth (femur) in relation to muscle length can result in a decrease in flexibility (hamstring) and strength (*De Ste Croix & Korff, 2013*). In addition, these muscles have been the most studied muscle group by scientific literature, given that they represent one of the most injured areas in sport, in general, and soccer, in particular (*Nogueira et al., 2017*; *Price et al., 2004*). Several published injury prevention programs include hamstring dynamic or static stretching exercises in order to increase the flexibility in these muscles (*DiStefano et al., 2009*; *Kiani et al., 2010*). Perhaps, the great incidence rates of hamstring injuries could explain an increase in the use of flexibility training as a hamstring injury prevention strategy by the practitioners, allowing better scores in HF-KE test in more trained athletes. Likewise, repetitive soccer skills as kicking the ball where a hip flexion is performed accompanied by a knee almost full extended could be a feasible reason to explain why older players present higher values in HF-KE ROM. These adaptations in HF-KE ROM will be again in agreement with the results shown by *López-Valenciano et al. (2019)* in professional soccer players (80.3° HF-KE ROM).

Appropriate body status requires a minimum of flexibility to respond to the demands of sport practise. Although there is no strong scientific evidence, flexibility appears to be also necessary for sport performance in youth. In U16 soccer players, greater flexibility of the lower back and upper thigh might discriminate players with high skill levels (*Vaeyens et al.,*

*2006*); in young soccer players with an age range from 14 to 18 year olds, hamstring flexibility was established as a key factor for performing football-specific skills, such as sprinting, jumping, agility, and kicking (*García-Pinillos et al., 2015*). The efficacy of the traditional stretching techniques (such as static, dynamic, PNF or ballistic) in the improvement of ROM results have been reported in several studies (*Ayala, Sainz de Baranda & Cejudo, 2012*; *Ayala, Sainz de Baranda & De Ste Croix, 2012*; *Ayala et al., 2013*; *Sainz de Baranda & Ayala, 2010*); by the same token, new protocols as the combination of traditional techniques with electrical stimulation (*Piqueras-Rodríguez, Palazón-Bru & Gil-Guillén, 2016*) have shown important enhancements in the flexibility of soccer players with reduced ROM. Therefore, the inclusion of sport-specific flexibility work in training routines could help to increase the ROM values and to improve sport performance. The appropriate stretching protocols and work-loads have to be selected by the practitioners to increase flexibility and to maintain it throughout the distinct stages of youth physical development, following the recommendations proposed by *Lloyd & Oliver (2012)*. Specially, these exercises should be aimed at young players with limited ROM; so it is essential to evaluate athletes' flexibility profile. On this matter, new predictive mobile apps have been published to determine boy soccer players with a higher muscle (hamstring) ROM restriction which can be used to make a simple and quickly assessment of our soccer teams (*Piqueras-Rodríguez et al., 2016*).

However, it is unclear what level of flexibility is optimum to improve performance and to prevent injuries in soccer. Optimal values may vary between muscle groups and different sports (*Ostojić & Stojanović, 2007*; *Sainz de Baranda et al., 2015*); while a swimmer needs higher values in the shoulder and ankle ROM, a dancer requires huge values in almost all (upper and lower extremity) ROM. Even within the same sport, the optimal values depend on the demands of the players' position and age; the goalkeeper will need distinct upper extremity ROM than the field player, and the differences in the movement patterns developed by children and adolescents will possibly require diverse degrees of ROM (as it happens with knee angular velocity in kicking (*Kellis & Katis, 2007*). The current study intends to analyse the flexibility differences between young soccer players in relation to their chronological age. Due to the lack of flexibility normative values in youth soccer, the flexibility profile derived from our results could be used as minimum objective scores to achieve in each age-group. Nonetheless, and having into account that soccer success is based on many different variables (*Nikolaidis et al., 2014*), it is necessary to publish prospective cohort studies adopting holistic approaches for predicting the exact cut off score in each physical skill that implies a higher sport performance and lower injury risk.

One of the principal limitations of this research was the sample size used in each group (U10, U12, U14, U16 and U19); the limited number of players in each of the youth academy teams hinders the athletes' involvement. Other limitation could be that neither index of maturity nor player position was assessed within the study. Future studies should analyse possible differences in flexibility in regard to players' maturational status and tactical position.

## CONCLUSIONS

U10 and U12 soccer players display highest values in five of the ROM assessed (HAD-HF90°, HAB, HIR, HER, HAB-HF90°). On the contrary, older players (U19) showed better ROM in HF-KE and HF-KF tests in comparison with their younger peers. The ROM reductions in internal and external hip rotation and hip adduction and abduction movements throughout the studied ages could explain an adaptive response of the muscles involved in playing soccer. The ROM restriction could predispose players to abnormal movement patterns and increase the risk of future injuries in the joints involved. Thus, the findings of this study reinforce the necessity of prescribing exercises aimed at improving HAD-HF90° ROM in U16, HAB ROM in U14, HIR ROM in U16 and U19, HER ROM in U12 and U19, and HAB-HF90° ROM in U16 and U19 players within everyday soccer training routines.

### Funding

This research was made possible by the Program of Human Resources Formation for Science and Technology grant number 20326/FPI/2017 from the Seneca Foundation-Agency for Science and Technology in the Region of Murcia (Spain). This study is part of the project entitled "Estudio del riesgo de lesión en jóvenes deportistas a través de redes de inteligencia artificial", funded by the Spanish Ministry of Science and Innovation (DEP2017-88775-P), the State Research Agency (AEI) and the European Regional Development Fund (ERDF). The funders had no role in study design, data collection and analysis, decision to publish, or preparation of the manuscript.

### Grant Disclosures

The following grant information was disclosed by the authors:
Program of Human Resources Formation for Science and Technology: 20326/FPI/2017.
Seneca Foundation-Agency for Science and Technology in the Region of Murcia (Spain).
Spanish Ministry of Science and Innovation: DEP2017-88775-P.

### Competing Interests

The authors declare there are no competing interests.

### Author Contributions

- Antonio Cejudo conceived and designed the experiments, performed the experiments, analyzed the data, prepared figures and/or tables, authored or reviewed drafts of the paper, approved the final draft.
- Francisco Javier Robles-Palazón performed the experiments, prepared figures and/or tables, authored or reviewed drafts of the paper, approved the final draft.
- Francisco Ayala performed the experiments, analyzed the data, contributed reagents/materials/analysis tools, approved the final draft.

- Mark De Ste Croix contributed reagents/materials/analysis tools, authored or reviewed drafts of the paper, approved the final draft.
- Enrique Ortega-Toro analyzed the data, contributed reagents/materials/analysis tools, approved the final draft.
- Fernando Santonja-Medina conceived and designed the experiments, contributed reagents/materials/analysis tools, authored or reviewed drafts of the paper, approved the final draft.
- Pilar Sainz de Baranda conceived and designed the experiments, contributed reagents/materials/analysis tools, prepared figures and/or tables, authored or reviewed drafts of the paper, approved the final draft.

### Human Ethics

The following information was supplied relating to ethical approvals (i.e., approving body and any reference numbers):

The experimental procedures used in this study were approved by the Ethics and Scientific Committee of the University of Murcia (Spain) (ID: 1551/2017).

### Data Availability

The raw data is provided in the Dataset 1.

### Supplemental Information

Supplemental information for this article can be found online at http://dx.doi.org/10.7717/peerj.6236#supplemental-information.

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
