# Peer review of "Age-related differences in flexibility in soccer players 8–19 years old"

_PeerJ, doi:10.7717/peerj.6236_

## Round 0.1 · original submission · Major Revisions

Dear authors,

Your manuscript has been reviewed by several experts in the analyzed topic and all of them have indicated scientific merit in your work. However, there are some issues which you must address in a revised version of the text. Furthermore, consider to reference:

1: Piqueras-Rodríguez F, Palazón-Bru A, Martínez-St John DR, Folgado-de la Rosa
DM, Gil-Guillén VF. A Tool to Quickly Detect Short Hamstring Syndrome in Boys who
Play Soccer. Int J Sports Med. 2016 Jan;37(1):1-5. doi: 10.1055/s-0035-1554699.
Epub 2015 Oct 28. PubMed PMID: 26509385.


2: Piqueras-Rodríguez F, Palazón-Bru A, Gil-Guillén VF. Effectiveness Analysis of
Active Stretching Versus Active Stretching Plus Low-Frequency Electrical
Stimulation in Children Who Play Soccer and Who Have the Short Hamstring
Syndrome. Clin J Sport Med. 2016 Jan;26(1):59-68. doi:
10.1097/JSM.0000000000000188. PubMed PMID: 25831408.

With respect and warm regards,
Dr Palazón-Bru (academic editor for PeerJ)


# ·

Basic reporting

no comment

Experimental design

no comment

Validity of the findings

no comment

Additional comments

GENERAL COMMENT
Considering the popularity of soccer, the importance of flexibility for health and the lack of studies on flexibility in soccer, this paper has practical relevance for coaches and trainers in soccer. I would recommend it for publication once the authors addressed a few minor issues.

SPECIFIC COMMENT
1. Abstract: “Muscle flexibility is one of the basic components of fitness for sport performance”. It must be revised here and throughout the text that flexibility is a main component of HEALTH-related fitness and is component of SPORT-related fitness only for those sports where it correlates with performance.
2. Abstract: Add basic characteristics of participants in methods.
3. Abstract: Add some mean and SD scores in the results.
4. Abstract: Add a sentence commenting on the evaluation of the flexibility in soccer compared to general population in results and conclusions.
5. Introduction: “Muscle flexibility is also one of the basic components of fitness for sport performance” should change. Flexibility is not related with sport performance in soccer, but with health.
6. Introduce: Before aims, add 1-2 leading sentences about general trend of changes in flexibility by age in the ages you plan to study.
7. Discussion: Although the literature on soccer and flexibility (Malina, Ribeiro, 71 Aroso, & Cumming, 2007; Manning & Hudson, 2009; Nikolaïdis, 2012; Vaeyens et al., 2006) is mentioned in the introduction, surprisingly it is not discussed. These references are major, thus, they should be discussed in 1-2 paragraphs after the 1st paragraph of discussion.
8. Discussion: Major literature is missing and should be discussed.
Inter-individual Variability in Soccer Players of Different Age Groups Playing Different Positions. Nikolaidis P, Ziv G, Lidor R, Arnon M. J Hum Kinet. 2014 Apr 9;40:213-25. doi: 10.2478/hukin-2014-0023.
Age- and sex-related differences in the anthropometry and neuromuscular fitness of competitive taekwondo athletes. Nikolaidis PT, Buśko K, Clemente FM, Tasiopoulos I, Knechtle B. Open Access J Sports Med. 2016 Dec 7;7:177-186
9. Fig 2 needs error bars.

Reviewer 2 ·

Basic reporting

Clear an unambiguous work

Literature well referenced & relevant.

Structure conforms to PeerJ standards, discipline norm

Figures are relevant, high quality, well labelled & described.(See details)

Experimental design

The investigation must have been conducted rigorously and to a high technical standard. The research must have been conducted in conformity with the prevailing ethical standards in the field.
Methods should be described with sufficient information to be reproducible by another investigator

Validity of the findings

Meaningful replication encouraged where rationale & benefit to literature is clearly
stated.

Data is robust, statistically sound, & controlled.

Speculation is welcome but should be identified in discusión (What is the reason through age, the decrease in flexibility ?)

Conclussion

There is a tendency to repeat results. Must be improved
Why is recommended reinforce the necessity of prescribing exercise?

Additional comments

1.Abstract. In results define before the acronims

2.- Fig 2: recommended put he SD in every point

Reviewer 3 ·

Basic reporting

The paper is well written

Experimental design

No comment

Validity of the findings

No comment

Additional comments

The aim of the study entitled “Age-related differences in flexibility in soccer players 8-19 years old” identify age-related differences in lower extremity flexibility in young soccer players. This manuscript contains confirmative/descriptive information.
The paper is well written. However there are same shortcomings in the statistical analysis, results and discussion sections that must be addressed in order to improve the quality of the paper.

Introduction
Line 50: The body composition is another important performance factor in soccer. Please provide an overview on why monitoring body composition is important for soccer players. The following recent citations should be added: (Milanese et al., 2015, Seasonal DXA-measured body composition changes in professional male soccer players; Carling and Orhant, 2010, Variation in body composition in professional soccer players: Interseasonal and intraseasonal changes and the effects of exposure time and player position).
Statistical analysis
How did you calculate the inter- and intra-operator variability in the measurement? The values of inter- and intra-operator variability must be reported.
Was the statistical power adequate? Please comment.
Results
In this section, the results about the ROM test for each leg were not reported in the text and in the table. Please clarify this statement.

Discussion
In this section, the results about the ROM test for each leg should be discussed.

---

## Round 0.2 · accepted · Accept

Dear authors,

I am pleased to report that your paper has been accepted for publication in PeerJ.

Congratulations!

With respect and kind regards,
Dr Palazón-Bru (academic editor for PeerJ)

·

Basic reporting

It's OK.

Experimental design

It's OK.

Validity of the findings

It's OK.

Additional comments

Great work! The paper has been improved a lot. I have no further comment.

Reviewer 2 ·

Basic reporting

All request has been done

Experimental design

All request has been done

Validity of the findings

All request has been done

Additional comments

All request has been done

Reviewer 3 ·

Basic reporting

No comment.

Experimental design

No comment.

Validity of the findings

No comment.

Additional comments

No comment.